# Identifying Temporal Orientation of Word Senses

### Abstract

The ability to capture the time information conveyed in natural language is essential to many natural language processing applications such as information retrieval, question answering, automatic summarization, targeted marketing, loan repayment forecasting, and understanding economic patterns. Therefore, a lexical temporal resource associating word senses with their underlying temporal orientation would be crucial for the computational tasks aiming at interpretation of language of time in text.

In this paper, we propose a semi-supervised minimum cuts paradigm that makes use of WordNet definitions or 'glosses', its conceptual-semantic and lexical relations to supplement WordNet entries with information on the temporality of its word senses. Intrinsic and extrinsic evaluation results show that the proposed approach outperforms prior semi-supervised, non-graph classification approaches to the temporality recognition of word senses/concepts, and confirm the soundness of the proposed approach.

## 1 Introduction

There is considerable academic and commercial interest in processing time information in text, where that information is expressed either explicitly, or implicitly, or connotatively. Recognizing such information and exploiting it for Information Retrieval (IR) and Natural Language Processing (NLP) tasks are important features that can significantly improve the functionality of NLP/IR applications.

Automatic identification of temporal expressions in text is usually performed either via time taggers (Strötgen and Gertz, 2013), which contain pattern files, such as uni-grams and bi-grams used to express temporal expressions in a given language (e.g. names of months) or various grammatical rules. As a rule-based system, time taggers are limited by the coverage of the rules for the different types of temporal expressions that it recognizes. To exemplify, the word *'present'* in the sentence *"Apple's iPhone is one of the most popular smartphones at present"* when labeled by SUTime[1] is tagged as:

*<TIMEX3      tid="t1"      type="DATE" value="PRESENT_REF">present</TIMEX3>*

It rightly tags the word *'present'* in the above example, and refers to it as the present time when reference date is considered as same as the tagging date. However, such word based indicators can be misleading. For example, below is the tag from SUTime for the word *'present'* in the sentence *"I was in Oxford Street getting the wife her birthday present"*. The tag gives us a false impression by wrongly labeling the word as a temporal one.

*<TIMEX3      tid="t1"      type="DATE" value="PRESENT_REF">present</TIMEX3>*

Reasons for this misleading information are i) time taggers usually do not use contextual indicators while deciding on temporality ii) different word senses of a single word can actually be either temporal or atemporal. A typical temporally-ambiguous word, i.e. a word that has at least one temporal and at least one atemporal sense, is 'present', as shown by the two examples above.

Whereas most of the prior computational linguistics and text mining temporal studies have focused on temporal expressions and events, there has been a lack of work looking at the tempo-

---

[1] http://nlp.stanford.edu:8080/sutime/process

ral orientation of word senses. Therefore, we focus our study on automatically time-tagging word senses into *past, present, future,* or *atemporal* using their WordNet (Miller, 1995) definition, instead of tagging temporal words.

In this paper, we put forward a semi-supervised graph-based classification paradigm build on an optimization theory namely the max-flow min-cut theorem (Papadimitriou and Steiglitz, 1998). In particular, we propose minimum cut in a connected graph to time-tag each synset of WordNet to one of the four dimensions: *atemporal*, *past*, *present,* and *future*. Our methodology was evaluated both intrinsically and extrinsically. It outperformed prior approaches to the temporality recognition of word senses/concepts. First, a gold standard is created using a crowdsourced annotation service to test our methodology. Second, temporal classification task is performed. Results show qualitative improvements when compared to previous state-of the-art approaches. Results also evidence that to achieve the performance of a standard supervised approach with our model, we need less than 10% of the training data.

## 2 Related Work

**Temporality in NLP and IR:** Temporality has recently received increased attention in Natural Language Processing (NLP) and Information Retrieval (IR). Initial works proposed in NLP and IR are exhaustively summarized in (Mani et al., 2005). The introduction of the TempEval task (Verhagen et al., 2009) and subsequent challenges (TempEval-2 and -3) in the Semantic Evaluation workshop series have clearly established the importance of time to deal with different NLP tasks. In IR, the work of (Baeza-Yates, 2005) defines the foundations of Temporal-IR. Since, research have been tackling several topics such as query understanding (Metzler et al., 2009), temporal snippets generation (Alonso et al., 2007), temporal ranking (Kanhabua et al., 2011), temporal clustering (Alonso et al., 2009), or future retrieval (Radinsky and Horvitz, 2013).

In order to push forward further research in temporal NLP and IR, (Dias et al., 2014) developed TempoWordNet (TWn), an extension of WordNet (Miller, 1995), where each synset is augmented with its temporal connotation (past, present, future, or atemporal). It mainly relies on the quantitative analysis of the glosses associated to synsets,

and on the use of the resulting vectorial term representations for semi-supervised synset classification. While (Hasanuzzaman et al., 2014a) show that TWn can be useful to time-tag web snippets, less comprehensive results are shown in (Filannino and Nenadic, 2014), where TWn learning features did not lead to any classification improvements. In order to propose a more reliable resource, (Hasanuzzaman et al., 2014b) recently defined two new propagation strategies: probabilistic and hybrid respectively leading to TWnP and TWnH. Although some improvements was evidenced, no conclusive remarks could be reached.

There are several disadvantages of their approaches. First, it relies mostly on WordNet glosses and do not effectively exploit WordNet's relation structure. Whereas we concentrate on the use of WordNet relations, glosses, and other attributes for the classification process. Second, strategies adopted to build TWnP and TWnH mainly depend on the probability estimates for the classes from Support Vector Machines (SVM) classifiers. However, probabilities are derived without using any post-calibration method. Converting scores to accurate probability estimates for multiclass problems requires a post-calibration procedure. In addition, there is no standard evaluation as to the accuracy of their approach.

**Graph-based Classification:** To the best of our knowledge, we present the first work, which aims to recognize temporal dimension of word senses via graph-based classification algorithm. However, graph-based algorithms have been used to classify sentences and documents into subjective/objective or positive/negative level (Pang and Lee, 2004; Agarwal and Bhattacharyya, 2005), instead of aiming at tagging at word sense level as we do. At the word level, a semi-supervised spin model is used for word polarity determination, where the graph is constructed using a variety of information such as gloss co-occurrences and WordNet links (Takamura et al., 2005). However, their model differs from ours.

## 3 Semi-supervised Mincuts

### 3.1 Minimum cuts: Main Idea

Underlying idea behind classification with minimum cuts (Mincuts) in graph is that similar items should be grouped together in the same cut. Suppose we have $n$ items $x_1, \ldots x_n$ to divide into two classes $C_1$ and $C_2$ based on the two types of in-

formation at hand. The first one, *individual* score $ind_j(x_i)$ is the non-negative estimate of each $x_i$'s preference for being in class $C_j$ based on the features of $x_i$ alone. While the later one, *association* scores $assoc(x_i, x_k)$ represent a non-negative estimates of how important it is that $x_i$ and $x_k$ be in the same class. Overall idea is to maximize each item's net score i.e. individual score for the class it is assigned to minus its individual score for the other class and penalize putting tightly associated items into different classes. It can be seen as the following optimization problem: assign the $x_i$s to $C_1$ and $C_2$ so as to minimize the partition cost.

$$\sum_{x \in C_1} ind_2(x) + \sum_{x \in C_2} ind_1(x) + \sum_{x_i \in C_1, x_k \in C_2} assoc(x_i, x_k) \quad (1)$$

We could represent the situation by building an undirected graph $G$ with vertices $\{v_1, \ldots, v_n, s, t\}$; $s$ and $t$ are *source* and *sink* respectively. Add $n$ edges $(s, v_i)$, each with weight $ind_1(x_i)$, $n$ edges $(v_i, t)$, each with weight $ind_2(x_i)$. Finally, add $\binom{n}{2}$ edges $(v_i, v_k)$, each with weight $assoc(x_i, x_k)$. Finally, cuts in $G$ are defined as follows:

**Definition 1.** *A cut $(S, T)$ of $G$ is a partition of its nodes into sets $S = \{s\} \cup S'$ and $T = \{t\} \cup T'$ where $s \notin S'$, $t \notin T'$. Its cost $cost(S, T)$ is the sum of the weights of all edges crossing from $S$ to $T$. A minimum cut of $G$ is one of minimum cost.*

Figure 1 illustrates an example of the concepts for classifying three items. Brackets enclose example values; here, the individual scores happen to be probabilities. Based on *individual* scores alone, we would put $Y$ ("Promise.") in $s$ (Temporal class), $N$ ("Chair") in $t$ (Atemporal class), and be undecided about $M$ ("Oath"). But the *association* scores favour cuts that put $Y$ and $M$ in the same class, as shown in the table. Thus, the minimum cut, indicated by the dashed red line, places $M$ together $Y$ in $s$.

### 3.2 Advantages

Formulating the task of temporality detection problem on word senses in terms of graphs allows us to model item-specific and pair-wise information independently. Therefore, it is a very flexible paradigm where we have two different views on the data. For example, rule based approaches or machine learning algorithms employing linguistic and other features representing temporal indicators can be used to derive *individual* scores for a particular item in isolation. The

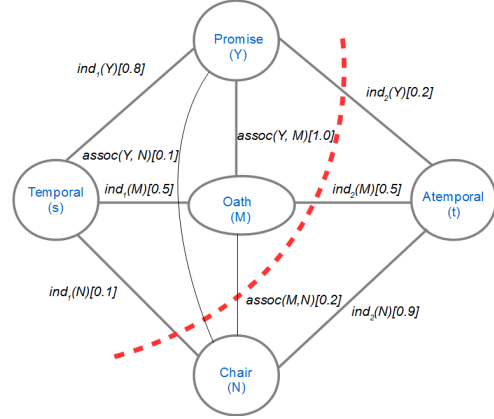

| s | Individual penalties | Association penalties | Cost |
|---|---|---|---|
| {Y,M} | .2+.5+.1 | .1+.2 | 1.1 |
| (none) | .8+.5+.1 | 0 | 1.4 |
| {Y,M,N} | .2+.5+.9 | 0 | 1.6 |
| {Y} | .2+.5+.1 | 1.0+.1 | 1.9 |
| {N} | .8+.5+.9 | .1+.2 | 2.5 |
| {M} | .8+.5+.1 | 1.0+.2 | 2.6 |
| {Y,N} | .2+.5+.9 | 1.0+.2 | 2.8 |
| {M,N} | .8+.5+.9 | 1.0+.1 | 3.3 |

Figure 1: Example graph.

edges weighted by the *individual* scores of a vertex (=word sense) to the source/sink can be interpretative as the probability of a word sense being temporal or atemporal without taking similarity to other senses into account. And we could also simultaneously use conceptual-semantic and lexical relations from WordNet to derive the *association* scores. The edges between two items weighted by the *association* scores can indicate how similar/different two senses are. If two senses are connected via a temporality-preserving relation they are likely to be both temporal or in opposite atemporal. An example here is the hyponymy relation, a temporality-preserving relation (Dias et al., 2014), where two hyponymy such as *present, nowadays—the period of time that is happening now* and *now—the momentary present* are both temporal. To detect the temporal orientation of word senses, authors in (Dias et al., 2014) adopted a single view instead of two on the data. The ability to combine two views on the data is precisely one of the strengths of our approach.

Second, Mincuts can be easily expanded into a semi-supervised framework. This is essential as the existing labeled datasets for our problem are small. In addition, glosses are short, leading to sparse high dimensional vectors in standard feature representations. Also, WordNet connections between different parts of the WordNet hierarchy can also be sparse, leading to relatively isolated senses in a graph in a supervised framework.

Semi-supervised Mincuts allow us to import unlabeled data that can serve as bridges to isolated components. More importantly, the unlabeled data can be related to the labeled data (by some WordNet relation), it might help pull unlabeled data to the right cuts (categories).

### 3.3 Formulation of Semi-Supervised Mincuts

Formulation of our semi-supervised Mincut for synset temporality classification involves the following steps.

I We depute two vertices $s$ (source) and $t$ (sink) which corresponds to the *"temporal"* and *"atemporal"* category respectively. We call the vertices $s$ and $t$ *classification vertices*, and all other vertices (labeled, test, and unlabeled data) *example vertices*. Each example vertex corresponds to one WordNet synset and is connected to both $s$ and $t$ through an undirected weighted edge. This confirms that the graph is connected.

II The labeled training examples are connected to classification vertices they belong to via edges with high constant non-negative weight. Unlabeled examples are connected to the classification vertices via edges weighted with non-negative scores that indicate the degree of association with temporal/atemporal category. We use a classifier to assign these edges weights.

III Conceptual-semantic and lexical relations available in the WordNet are used to construct edges between two example vertices. Such edges can exist between any pair of example vertices, for instance between two unlabeled vertices.

IV After building graph, we then employ a maximum-flow algorithm to find the minimum $s - t$ cuts of the graph. The cut in which the source vertex $s$ lies is classified as *"temporal"* and the cut in which sink vertex $t$ lies is labeled as *"atemporal"*.

V In order to fine tune the temporal part, we follow hierarchical strategy by organizing the classes (past, present, future) according to a hierarchy. The hierarchy of classes is decided based on the classes that are easier to discriminate to improve the overall classification accuracy. First, we define two vertices $s$ (source) and $t$ (sink) which correspond to the

*"past"* and *"Not_Past"* temporal categories respectively. Then we follow the above steps $I$ through $IV$. This divides the temporal part into two disjoint subsets: *past* synsets and synsets belong to *present* and *future* temporal category. Finally, we repeat steps $I$ through $IV$ where vertices $s$ (source) and $t$ (sink) corresponds to *"future"* and *"present"* category respectively.

**Labeled and unlabeled data selection:** We use the same *temporal (past, present, future)* and *atemporal* sets of synsets considered as training data at the time of building TempoWordNet (TWnL) (Dias et al., 2014) as training /labeled data for our experiments. For test set, sample of synsets outside the labeled data is selected and annotated using crowdsourcing service. All other synsets outside labeled and test set are considered as unlabeled data.

**Weighting of edges to the classification vertices:** The edge weight (non-negative) to the source $s$ and the sink $t$ denotes how likely it is that an example vertex is put in the cut in which $s$ (*temporal*) or $t$ (*atemporal*) lies. For the unlabeled and test examples, a supervised learning strategy (over the labeled data as training set) is used to assign the edge weights. Each synset is represented by its gloss encoded as a vector of word unigrams weighted by their frequency in the gloss. As for classifier, we used SVM from the Weka platform[2]. In order to ensure that Mincut does not reverse the labels of the labeled training data, we assign a high[3] constant non-negative weight of 3 to the edge between a labeled vertex and its corresponding classification vertex, and a low weight of 0.001 to the edge to the other classification vertex.

**Deriving weights for WordNet relations:** While formulating the graph, we connect two vertices by an edge if they are linked by one of the ten (10) WordNet relations in Table 1. Main motivation towards using other relations in addition to the most frequently encoded relations (hypernym, hyponym) among synsets in WordNet is to achieve high graph connectivity. Moreover, we can assign different weights to different relations to reflect the degree to which they are temporality preserving. Therefore, we adopt two strategies to assign

---

[2]http://www.cs.waikato.ac.nz/ml/weka/ [Last access: 12/04/2015].

[3]w.r.t. the probability estimates (after calibration) of the classes from SVM.

weights to different WordNet relations. The first method (*ScWt*), assigns the same constant weight of 1.0 to all WordNet relations.

The second method (*DiffWt*), considers several degrees of preserving temporality. In order to do this, we adopt a simple strategy to produce large noisy set of *temporal* [4] and *atemporal* synsets from WordNet. This method uses a list of 30 hand-crafted temporal seeds (equally distributed over *past, present*, and *future* temporal categories) proposed in (Dias et al., 2014) along with their direct hyponym[5] to classify each WordNet synset with at least one temporal word[6] in its gloss as temporal and all other synsets as atemporal. We then simply count how often two synsets connected by a given relation (edge) have the same or different temporal dimension. Finally, weight is calculated by #same/(#same+#different). Results are reported in Table 1.

| Wordnet Relation | #same | #different | Weight |
|------------------|-------|-----------|--------|
| Direct-Hypernym | 61914 | 9600 | 0.76 |
| Direct-Hyponym | 73268 | 7246 | 0.91 |
| Antonym | 1905 | 3614 | 0.35 |
| Similar-to | 6587 | 1914 | 0.77 |
| Derived-from | 3630 | 1947 | 0.65 |
| Also-see | 1037 | 337 | 0.75 |
| Attribute | 350 | 109 | 0.76 |
| Troponym | 6917 | 2651 | 0.72 |
| Domain | 2380 | 2895 | 0.45 |
| Domain-member | 2380 | 2895 | 0.45 |

Table 1: WordNet relation weights (DiffWt Method)

## 4 Experiments and Evaluation

### 4.1 Datasets

**Labeled Data:** We used a list that consists of 632 *temporal* synsets in WordNet and equal number of *atemporal* synsets provided by (Dias et al., 2014) as labeled data for our experiments. Temporal synsets are distributed as follows: 210 synsets marked as *past*, 291 as *present*, and 131 as *future*.

**Building of a Gold Standard:** Since to our best knowledge, there is no gold standard resource with temporal association of words (except 30 hand-crafted temporal seeds proposed by Dias et al., 2014), we designed our own annotation task using the crowdsourcing service of CrowdFlower platform[7]. For the annotation task, three hundred

ninety eight (398) synsets equally distributed over nouns, verbs, adjectives and adverbs POS categories along with their lemmas and glosses are selected randomly from WordNet [8] as representative of the whole WordNet. Note that this number of synset is statistically significant representative sample of total WordNet synsets (117000 plus synsets) and derived using the formula described in (Israel, 1992). Afterwards, we designed two question that the annotators were expected to answer for a given synset (lemmas and gloss are also provided). While the first question is related to the decision which reflects a synset being *temporal* or *atemporal*, the motivation behind the second question is to collect a more fine-grained (*past, present, future*) **gold-standard** for synset-temporality association. Details of annotation guideline is out of scope of this paper.

The reliability of the annotators was evaluated on the basis of 60 control synsets [9] provided by (Dias et al., 2014) which were clearly associated either with a specific *temporal* or *atemporal* dimension and 10 temporally ambiguous synsets associated with more than one *temporal* dimensions. Similar to (Tekiroğlu et al., ), the raters who scored at least 70% accuracy on average on both sets of control synsets were considered to be reliable. Each unit was annotated by at least 10 reliable raters.

Table 2 demonstrates the observed agreement. Similar to (Mohammad, 2011; Özbal et al., 2011), annotations with a majority class greater than 5 is considered as reliable. Indeed for *temporal vs atemporal*) classification, 84.83 % of the synset annotations the absolute majority agreed on the same decision, while for *past, present,* and *future*, 72.36% of the annotations have majority class greater than 5. The high agreement confirms the quality of the resulting gold standard data.

### 4.2 Semi-supervised Graph Mincuts

**Temporal Vs Atemporal Classification:** Using our formulation in Section 3.3, we construct a connected graph by importing 1264 training set (632 *temporal* and 632 *atemporal* synsets), 398 gold standard test set created using a crowdsourced service, and 115996 unlabeled synsets[10]. We con-

---

[4] To fine tune the temporal part we used the same strategy for tagging past, present, and future synsets following hierarchical strategy

[5] Relation which preserves temporality according to (Dias et al., 2014)

[6] Most frequent sense of the temporal word from WordNet is selected

[7] http://www.crowdflower.com/

[8] WordNet version 3.0 used and selected outside from the labeled data set

[9] 30 temporal synsets (equally distributed over past, present, future) and 30 atemporal synsets

[10] All synset of WordNet−(training set+test set)

| Majority Class | 3 | 4 | 5 | 6 | 7 | 8 | 9 | 10 |
|---|---|---|---|---|---|---|---|---|
| *Synset as temporal or atemporal* | 0 .20 | 1.21 | 4.32 | 10.69 | 14.56 | 29.34 | 19.23 | 11.01 |
| *Temporal synset into past, present, or future* | 1.23 | 3.01 | 10.45 | 20.22 | 16.56 | 12.34 | 14.23 | 9.01 |

Table 2: Percentage of synsets in each majority class.

struct edge weights to *classification vertices, s (temporal)* and *t (atemporal)* by using a SVM classifier discussed above. WordNet relations for links between *example vertices* are weighted either by non-negative constant value (*ScWt*) or by the method *'DiffWt'* illustrated in Table 1.

**Temporally tagged synsets into past, present, and future:** In order to fine tune the temporal part, we construct another connected graph by importing 632 labeled temporal synsets, temporal part of the gold standard test data (127 synsets out of 398 synsets), temporally tagged synsets as unlabeled data [11] and follow the same strategy presented in Section 3.3 where *classification vertices, s* and *t* correspond to *'past'* and *'Not_past'* temporal categories respectively. Finally, temporal synsets tagged as *'Not_past'* classified either as *present* or *future* following the same analogy.

### 4.3 Evaluation

The underlying idea being that a reliable resource must evidence high quality time-tagging as well as improved performance for some external tasks.

### 4.4 Intrinsic Evaluation

**Baseline:** In order to compare our semi-supervised Mincut approach to a reasonable baseline, we use rule based approach to classify gold-standard data into *past, present, future,* or *atemporal* based on its lemmas and glosses. First, time expressions in the glosses of synsets are labeled and resolved via Standford's SUTime tagger, which give accuracy in line with the state-of-the-art systems at identifying time expressions at TempEval. For each synset, Named-entity time tag provided by SUTime (e.g."future_ref" or "present_ref" etc.) for the time expression present in its gloss is considered as the temporal class for that particular synset. In case of more than one temporal expression present [12] in the gloss of a synset, majority class of the time tags is se-

lected. Secondly, if no time expression is identified by the time tagger, a list composed of 30 hand-crafted temporal seeds proposed in (Dias et al., 2014) along with their direct hyponym and standard temporal adverbials (since), prepositions (before/after), adjectives (former) etc. is used to classify synsets with at least one temporal word [13] in its lemma(s) and gloss as temporal (past, present, future) and all other synsets as atemporal. Finally, performance of the rule based approach is measured for the gold standard data set and presented in Table 3. To figure out the contribution of word sense disambiguation, classic Lesk algorithm (Lesk, 1986) is used to choose right sense/synset for a word instead of most frequent sense. We found that contribution is negligible (< 0.4% improvement in overall accuracy).

To strengthen comparative evaluation of our semi-supervised Mincut approach, we propose to test our methodology with prior works (TempoWordNet: TWnL, TWnP, and TWnH). Comparative evaluation results are presented in Table 3. Results show the Mincut approach (*CFG2*) outperforms state-of-the-art approaches. It achieves highest accuracies for both *temporal vs. atemporal* and *past, present, future* classification with improvement of **11.3**% and **10.3**% respectively over the second best TempoWordNet versions (TWnH). Considering the above findings, we select our best Mincut configuration *CFG2* for the remaining experiments. Distribution of time-tag synsets produced by this configuration: atemporal=110002, past=1733, present=4193, future=1730. Some examples are given below:

- *late–having died recently. (**Past**)*
- *present, nowadays–the period of time that is happening now. (**Present**)*
- *promise–a verbal commitment by one person to another. (**Future**)*
- *field–a piece of land cleared of trees and usually enclosed. (**Atemporal**)*

**Performance with different size training data:** We randomly generate subsets of labeled

---

[11] Total number of synsets classified as temporal − (Total number of temporal synsets in training data+Total number of gold standard temporal synsets tagged as temporal at the time of temporal vs atemporal classification process)

[12] Found very rarely < 1%

---

[13] Most frequent sense of the temporal word from WordNet is selected

Table 3: Accuracy (percentage classified correct) for *temporal vs. atemporal* and temporal into *past, present, future* classification using different methods namely TempoWordNet (TWnL, TWnP, TWnH) and Mincuts measured over created gold-standard data. *CFG1* corresponds to the Mincut that uses SVM classifier to infer edges weights of unlabeled and test examples to the classification vertices *s* and *t* and predefined constant weights for WordNet relations (*ScWt*). *CFG2* corresponds to the Mincut approach that uses the same SVM classifier to infer edges weights but uses a list of temporal synsets to infer weights of Wordnet relations (*DiffWt*). Both of our configurations perform significantly better than previous approaches. Results are also broken down by precision (p), recall (r), and f1 score for past, present, future, and atemporal categories.

| Method | Baseline | TWnL | TWnP | TWnH | CFG1 | CFG2 |
|---|---|---|---|---|---|---|
| **Accuracy** | 48.8 | 65.6 | 62.0 | 68.4 | 74.4 | **79.7** |
| temporal (p, r, f1) | (52.0, 56.3, 54.0) | (63.5, 82.1, 71.6) | (55.8, 84.2, 67.1) | (67.4, 81.9, 73.9) | (84.5, 79.8, 82.0) | (89.1, 79.3, 83.9) |
| atemporal (p, r, f1) | (58.2, 54.2, 56.1) | (68.3, 79.2, 73.3) | (58.9, 75.6, 66.2) | (69.3, 82.6, 75.3) | (81.3, 86.6, 83.8 ) | (87.4, 90.8, 89.1) |
| **Accuracy** | 45.6 | 62.0 | 59.6 | 65.7 | 72.7 | **76.0** |
| past (p, r, f1) | (49.3, 46.7, 47.9) | (61.2, 73.0, 66.5) | (59.3, 79.1, 67.7) | (63.1, 75.0, 68.0) | (71.1, 79.5, 75.0) | (81.2, 78.5, 79.8) |
| present (p, r, f1) | (55.3, 48.2, 51.5) | (63.0, 75.2, 68.5) | (58.0, 78.2 66.0 ) | (77.4, 69.2, 73.0) | (73.0, 71.5, 72.2) | (85.1, 74.7, 79.0) |
| future (p, r, f1) | (48.5, 49.0, 48.7) | (62.1, 71.9, 66.6) | (57.0, 83.1, 67.6) | (60.0, 75.6, 66.8) | (79.4, 69.5, 74.0) | (86.1, 70.0, 77.2) |

data/training data (1064 synsets: 632 *temporal* and 632 *atemporal*) $L_1, L_2, L_3......L_n$, and ensures that $L_1 \subset L_2 \subset L_3...... \subset L_n$. As proposed in (Dias et al., 2014), binary classification models based on the generated subsets of labeled data are learned. Using the same subsets of labeled data, we formulate our best performing minimum cut (*CFG2*). Accuracies of both approaches are presented in Table 4. As can be seen from the table, semi-supervised Mincut performs consistently better than the previous semi-supervised non-graph classification approach (SVM). Moreover, our proposed graph classification framework with only 400 labeled data/training data examples achieves even higher accuracies than SVM with 1264 training items *(73.7% vs 68.4% )*.

| Number of labeled data | SVM (TWnH) | Minncut (CFG2) |
|---|---|---|
| 100 | 59.8 | 64.3 |
| 200 | 62.6 | 67.5 |
| 400 | 65.5 | 73.7 |
| 600 | 67.4 | 77.6 |
| 800 | 67.9 | 79.2 |
| 1000 | 68.0 | 79.0 |
| 1264 (all) | 68.4 | 79.7 |

Table 4: Accuracy with different sizes of label data for temporal vs. atemporal classification.

### 4.5 Extrinsic Evaluation

For extrinsic evaluation, we focus on the problem of classifying temporal relations task of TempEval-3, assuming that identification of events, times are already performed. The underlying idea is that proposed method to yield a time-enhanced WordNet has a greater positive impact on more applied temporal information extraction tasks, whose output is useful for many information retrieval applications.

In order to produce comparative results with best performing system at TempEval-3 namely *UTTime* (Laokulrat et al., 2013) for the above task, we follow the guidelines and use the same data sets provided by the evaluation campaign organizers. We restrict[14] our experiment to a subset of relations namely *BEFORE (CORR. Past), AFTER (CORR. Future)*, and *INCLUDES* (CORR. Present) with all other relation mapped to the 'NA-RELATION' for *event to document creation time* and *event to same sentence event*. For the task, we adopt a very simple strategy and implement the following features.

- String features: The tokens and lemmas of each entity pair.
- Grammatical features: The part-of-speech (PoS) tags of the entity pair (only for event-event pairs). The grammatical information is obtained using the Standford CoreNLPtool[15].
- Entity attributes: Entity pair attributes as provided in the data set.
- Dependency relation: We used the information related to the dependency relation between two entities such as type of dependency, dependency order.
- Textual context: The textual order of entity pair.
- Lexica: The relative frequency of temporal categories, based on the resource developed in this research, in the text appearing between entity pair (event to same sentence event), text of all tokens in the time expression, 5 tokens following and preceding time expression/event. The features are encoded as the frequency with which a word from a temporal category (*past, present, future*) appeared in the text divided by the total number of tokens in the text.

---

[14] Because of the complexity to map 14 relations of TempEval-3 into three temporal classes (past, present, future) considered for this experiment

[15] http://stanfordnlp.github.io/CoreNLP/

We build our system namely $TRel_{Mincuts}$ (composed of two classifiers) using Support Vector Machine (SVM) implementation of Weka over the features set and training data provided by the organizers. The best classifier for *event to document creation time* and *event to same sentence event* relation are selected via a grid search over parameter settings. The grid is evaluated with a 5-fold cross validation on the training data. We also measure the performance of *UTTime* for the above stated settings. Additionally, we build $TRel_{TWnH}$ by adopting same strategy and features set *'Lexica'* is computed from prior time-tagging approach namely TWnH. Table 5 presents comparative evaluation results.

| Approaches | Precision(%) | Recall(%) | F1(%) |
|---|---|---|---|
| $UTTime$ | 57.5 | 58.7 | 58.1 |
| $TRel_{Mincuts}$ | **66.9** | **68.7** | **67.7** |
| $TRel_{TWnH}$ | 61.2 | 62.5 | 61.8 |

Table 5: Performance of different approaches on temporal relation classification based on TempEval-3 Evaluation strategy.

Results evidence that $TRel_{Mincuts}$ significantly outperforms all other approaches and achieve highest performance in terms of precision (**+5.7**), recall (**+6.2**), and $F1$ score (**+5.9**). Results also demonstrate that our approach achieves **9.6% improvement in terms of $F1$ score over the best performing system in TempEval-3**.

We perform feature ablation analyses as presented in Table 6 in order to measure features contribution. As can be seen, every features set produced improvement over baseline (*mfc*) and highest improvement is achieved with features set (*Lexica*) that comes from the proposed temporal lexical resource. The result also implies that while each feature type contains certain temporal information, there is also some redundancy across the feature types.

| Features | F1(%) | Features | F1(%) |
|---|---|---|---|
| mfc baseline | 33.55 | all features | **67.7** |
| *string* alone | 45.06 | w/o *string* | 65.70 |
| *grammatical* alone | 46.96 | w/o *grammatical* | 64.85 |
| *entity* alone | 52.23 | w/o *entity* | 62.08 |
| *dependency* alone | 48.65 | w/o *dependency* | 65.06 |
| *textual* alone | 46.82 | w/o *textual* | 64.96 |
| *lexica* alone | 51.62 | w/o *lexica* | 62.76 |

Table 6: Feature ablation analysis of $F1$ score. The most frequent class baseline (mfc) indicates accuracy if only predicting the present class.

## 5 Discussion

One important remark can be made related to the difficulty of the task. This is particularly due to the fact that the temporal dimension of synsets is mainly judged upon their definition. For example, "dinosaur" can be classified as temporal or atemporal as its gloss *"any of numerous extinct terrestrial reptiles of the Mesozoic era"* allows both interpretations. Apart from it, while digging into the results we observed that classifying the temporal synsets into *past, present,* or *future* is more difficult than *temporal vs. atemporal classification*. It is due to the fact that *past, present* and *future* connotations are only indicative of the temporal orientation of the synset but cannot be taken as a strict class. Indeed, there are many temporal synsets, which are neither past, present nor future (e.g.*monthly – a periodical that is published every month*).

## 6 Conclusions

In this paper, we proposed a semi-supervised minimum cut framework to address the relatively unexplored problem of associating word senses with their underlying temporal dimensions. The basic idea is that instead of using single view, multiple views on the data would result in better temporal classification accuracy thus lead to a accurate and reliable temporal lexical resource. Thorough and comparative evaluations are performed to measure the quality of the resource. The results confirm the soundness of the proposed approach and the usefulness of the resource for temporal relation classification task. The resource is publicly available on `https://www.anonymous.anonymous` so that the community can benefit from it for relevant tasks and applications.

From a resource point of view, we would like to explore the effect of other graph construction methods, such as the use of freely available online dictionaries including thesaurus and distributional similarity measures. We would also like to use the resource for various applicative scenarios such as automatic analysis of time-oriented clinical narratives. As an example, one can imagine a system that automatically analyzes medical discharge summaries including previous diseases related to the current conditions, treatments, and the family history for medical decision making, data modeling and biomedical research.

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
