# Peer review of "Identifying Temporal Orientation of Word Senses"

_CoNLL 2016 — decision unknown_

[Official Review · Reviewer 1 · rating 4 · confidence 2]
soundness 4 · originality 3 · clarity 4 · impact 4 · substance 4 · appropriateness 5 · meaningful comparison 5 · replicability 4 · presentation format Oral Presentation

This paper presents an approach to tag word senses with temporal information
(past, present, future or atemporal). They model the problem using a
graph-based semi-supervised classification algorithm that allows to combine
item specific information - such as the presence of some temporal indicators in
the glosses - and the structure of Wordnet - that is semantic relations between
synsets â, and to take into account unlabeled data. They perform a full
annotation of Wordnet, based on a set of training data labeled in a previous
work and using the rest of Wordnet as unlabeled data. Specifically, they take
advantage of the structure of the label set by breaking the task into a binary
formulation (temporal vs atemporal), then using the data labeled as temporal to
perform a finer grained tagging (past, present or future). In order to
intrinsically evaluate their approach, they annotate a subset of synsets in
Wordnet using crowd-sourcing. They compare their system to the results obtained
by a state-of-the-art time tagger (Stanford's SUTime) using an heuristic as a
backup strategy, and to previous works. They obtain improvements around 11% in
accuracy, and show that their approach allows performance higher than previous
systems using only 400 labeled data. Finally, they perform an evaluation of
their resource on an existing task (TempEval-3) and show improvements of about
10% in F1 on 4 labels.

This paper is well-constructed and generally clear, the approach seems sound
and well justified. This work led to the development of a resource with fine
grained temporal information at the word sense level that would be made
available and could be used to improve various NLP tasks. I have a few remarks,
especially concerning the settings of the experiments.

I think that more information should be given on the task performed in the
extrinsic evaluation section. An example could be useful to understand what the
system is trying to predict (the features describe âentity pairsâ but it
has not been made clear before what are these pairs) and what are the features
(especially, what are the entity attributes? What is the POS for a pair, is it
one dimension or two? Are the lemmas obtained automatically?). The sentence
describing the labels used is confusing, I'm not sure to understand what
âevent to document creation timeâ and âevent to same sentence eventâ
means, are they the kind of pairs considered? Are they relations (as they are
described as relation at the beginning of p.8)? I find unclear the footnote
about the 14 relations: why the other relations have to be ignored, what makes
a mapping too âcomplexâ? Also, are the scores macro or micro averaged?
Finally, the ablation study seems to indicate a possible redundancy between
Lexica and Entity with quite close scores, any clue about this behavior?

I have also some questions about the use of the SVM.  For the extrinsic
evaluation, the authors say that they optimized the parameters of the
algorithm: what are these parameters?  And since a SVM is also used within the
MinCut framework, is it optimized and how? Finally, if it's the LibSVM library
that is used (Weka wrapper), I think a reference to LibSVM should be included. 

Other remarks:
- It would be interesting to have the number of examples per label in the gold
data, the figures are given for coarse grained labels (127 temporal vs 271
atemporal), but not for the finer grained.
- It would also be nice to have an idea of the number of words that are
ambiguous at the temporal level, words like âpresentâ.
- It is said in the caption of the table 3 that the results presented are
âsignificantly betterâ but no significancy test is indicated, neither any
p-value.

Minor remarks:
- Related work: what kind of task was performed in (Filannino and Nenadic,
2014)?
- Related work: ârequires a post-calibration procedureâ, needs a reference
(and p.4 in 3.3 footnote it would be clearer to explain calibration)
- Related work: âtheir model differ from oursâ, in what?
- Table 3 is really too small: maybe, remove the parenthesis, put the
â(p,r,f1)â in the caption and give only two scores, e.g. prec and f1. The
caption should also be reduced.
- Information in table 4 would be better represented using a graph.
- Beginning of p.7: 1064 â 1264
- TempEval-3: reference ?
- table 6: would be made clearer by ordering the scores for one column
- p.5, paragraph 3: atemporal) â atemporal